# Merkel Cell Polyomavirus Is Associated with Anal Infections in Men Who Have Sex with Men

**DOI:** 10.3390/microorganisms7020054

**Published:** 2019-02-19

**Authors:** Nunzia Zanotta, Serena Delbue, Lucia Signorini, Sonia Villani, Sarah D’Alessandro, Giuseppina Campisciano, Claudia Colli, Francesco De Seta, Pasquale Ferrante, Manola Comar

**Affiliations:** 1Institute for Maternal and Child Health—IRCCS “Burlo Garofolo”, 34137 Trieste, Italy; zanottanunzia@gmail.com (N.Z.); giusi.campisciano@burlo.trieste.it (G.C.); Francesco.deseta@burlo.trieste.it (F.D.S.); manola.comar@burlo.trieste.it (M.C.); 2Department of Biomedical, Surgical and Dental Sciences, University of Milano, 20133 Milano, Italy; serena.delbue@unimi.it (S.D.); lucia.signorini@unimi.it (L.S.); sonia.villani@unimi.it (S.V.); sarah.dalessandro@unimi.it (S.D.); 3MST Center, ASUITS, 34100 Trieste, Italy; claudia.colli@asuits.sanita.fvg.it; 4Department of Medical Science, University of Trieste, 34127 Trieste, Italy

**Keywords:** men who have sex with men (MSM), sexually transmitted infection (STI), Merkel cell polyomavirus (MCPyV)

## Abstract

Background: Viral infections of the anal/rectal tract of men who have sex with men (MSM) have been poorly studied. Methods: In total, 158 swab samples (81 anal/rectal, 65 throat/oral and 12 urethral) were collected from 126 MSM. DNA was isolated and subjected to real-time PCR assays for the detection of the sexually transmitted (ST) pathogens *Chlamydia trachomatis*, *Neisseria gonorrhoeae* and *Mycoplasmas ssp*, human papillomavirus (HPV) and six human polyomaviruses (HPyVs; JCPyV, BKPyV, Merkel cell PyV–MCPyV-, HPyV-6, HPyV-7 and HPyV-9). Results: *C. trachomatis* (31/126, 24.6%) and *M. genitalium* (30/126, 23.8%) were the most frequently detected ST pathogens. Thirty-one/126 (24.6%) patients were positive for at least one HPyV. The significantly (*p* < 0.05) prevalent HPyV in the anal tract was MCPyV, which was amplified in 27/81 (33.3%) samples, followed by HPyV-6, which was amplified in 6/81 (7.4%) swabs. Coinfections with MCPyV and *C. trachomatis* or *Mycoplasmas* were found in 4/21 (19.0%) and 5/21 (23.8%) anal/rectal swabs, respectively. Three/4 MCPyV-*C. trachomatis* coinfected patients were symptomatic. Conclusions: Based on the high prevalence of MCPyV in the anal/rectal swabs from MSM patients and on the well-known oncogenic properties of MCPyV, sexual transmission and possible involvement of HPyVs in the pathogenesis of diseases of the anal canal should be further studied.

## 1. Introduction

Polyomaviruses (PyVs) are nonenveloped, double-stranded small DNA viruses with an icosahedral capsid [1]. Over the last 10 years, the number of identified species-specific PyVs has grown; in particular, 14 have been found in human tissues or specimens, including the well-known JC polyomavirus (JCPyV), BK polyomavirus (BKPyV) and Merkel cell polyomavirus (MCPyV) and the three new human polyomaviruses (HPyVs), human polyomavirus-6 and -7 (HPyV-6, -7), which, like MCPyV, commonly inhabit healthy human skin, and -9 (HPyV-9), which has been isolated from the serum of kidney transplant patients [2,3,4,5,6,7]. Most HPyV infections are common, with seropositivity values ranging from 60% to 90% in the general adult population, depending on the virus [8]. After the asymptomatic primary infection, a life-long persistence is established in different anatomical sites, such as lymphoid tissue (JCPyV), renal epithelium (JCPyV and BKPyV), and skin (MCPyV, HPyV-6 and -7) [9]. HPyV-associated diseases typically occur in patients with a deficient immune system [10]. Transmission is mostly by direct person-to-person contact and occurs in early childhood [11]. Detection of HPyVs in tonsils, prostate and kidney tissues as well as in urine and stool are well described in the literature, confirming the urino-oral and feco-oral routes of transmission [12]. Furthermore, Comar et al. and Rotondo et al. reported evidence of the presence of HPyVs, particularly the high prevalence of JCPyV in semen, suggesting the sexual route as a possible alternative for interhuman HPyV transmission [13,14].

Only two manuscripts have focused on the prevalence of HPyV infection in anal samples from men who had sex with men (MSM), which demonstrated that MCPyV was present in approximately 30% of the analyzed specimens [15].

Based on these limited results and on the severity of the diseases caused by HPyVs, we aimed to describe the prevalence of six HPyVs (JCPyV, BKPyV, MCPyV, HPyV-6, -7, and -9) in oral, anal/rectal and urethral samples from a cohort of MSM and to evaluate the presence of coinfection with other sexually transmitted (ST) pathogens responsible for ST infections (STIs).

## 2. Materials and Methods

### 2.1. Subjects’ Enrollment and Sample Collection

The cross-sectional study was conducted retrospectively on a total of 158 samples; throat/oral (65), anal/rectal (81) and urethral (12) swab samples were collected from 126 MSM with a mean age of 34.8 ± 10.9 that were negative for human immunodeficiency virus (HIV) infection, and at risk for STI. Symptomatic patients showed pruritic and dyskeratotic dermatitis of the anal area, proctitis and pelvic pain. For 29 subjects, both throat and anal swabs were available.

Samples were sent to the Advanced Translational Microbiological Department, Burlo Garofolo Institute, in Trieste (Italy) for molecular diagnosis of the STIs. Part of the DNA was used for subsequent HPyV screening. The study was approved by the local Ethics Committee, and all patients signed a free and informed consent form during the clinical examination.

### 2.2. Nucleic Acid Extraction

The swabs were diluted in 1 mL of physiological solution and were vortexed. Genomic DNA was extracted from 500 μL using the NucliSENS^®^ EasyMAG (Biomérieux S.p.a. Florence, Italy) automated system, according to the manufacturer’s instructions. All DNA samples were eluted in a final volume of 100 μL and stored at −80 °C prior to further processing.

### 2.3. STI Detection

Chlamydia trachomatis (*C. trachomatis*), *Neisseria gonorrhoeae* (*N. gonorrhoeae*), *Mycoplasma genitalium*/*hominis* and *Ureaplasma urealyticum*/*parvum* (*Mycoplasmas ssp.*). DNA was explored from the samples of participants using a multiplex Real Time PCR following the manufacturer’s instructions (Dia. Pro, Italy). HPV detection and genotyping were performed using the Anyplex™ II HPV Detection assay (Seegene Inc., Arrow diagnostics, Italy), as previously described [16].

### 2.4. Quantitative Real-Time PCR Assay (Q-PCR) for HPyVs

Each DNA sample was tested for the presence of six HPyVs: JCPyV, BKPyV, MCPyV, HPyV-6, HPyV-7, and HPyV-9. Q-PCR assays were performed using the Applied Biosystems 7500 Real-Time PCR System (Applied Biosystems, Foster City, CA, USA) with specific primers and TaqMan probe technology, targeting the Viral Protein 1 (VP1) sequence of BKPyV, MCPyV, HPyV-6, HPyV-7 and HPyV-9 and the Large T Antigen (LT) sequence of JCPyV (Table 1). In particular, MCPyV/BKPyV, JCPyV/HPyV6, HPyV7/HPyV9 duplex Q-PCRs were run in accordance with previously published protocols [17].

Briefly, each reaction contained 1× Universal Taqman PCR Master Mix (Solis BioDyne, Estonia), 5 µL of DNA and a variable amount of forward and reverse primers and probes, as summarized in Table 1. The thermal cycles were as follows: 50 °C for 2 min, 95 °C for 15 min followed by 40 cycles of 95 °C for 15 s and 60 °C for 1 min. Standard curves were constructed using a ten-fold dilution series of plasmids containing the entire VP1 gene of BKPyV, MCPyV, HPyV-6, HPyV-7 and HPyV-9 and the entire LT of JCPyV (dilution range: 10^8^–10 copies/μL). The limit of detection of each assay was 10 copies/μL. To determine the quality and percentage of infected cells, a concomitant Q-PCR assay targeting the β-globin gene was performed on the same samples using the primer set and thermal cycles previously published [18]. Negative and positive controls were included in each run. Each sample, standard and control, was tested in triplicate. The results were analyzed by the absolute quantification method and reported as copies/mL and percentage of infected cells, calculated as follows: ([viral copies]/[beta-globin copies/2]) × 100.

### 2.5. PCR Assays and Sequencing Analysis for MCPyV Molecular Characterization

MCPyV-positive DNA samples were subjected to two different PCR protocols for the amplification of two fragments belonging to the LT gene. The first set of primers, the outer TAg FwE and TAg RvE, and the inner TAg FwI and Tag RvI, amplify a 183-bp fragment [19]. The second seminested PCR was conducted using the outer primer set LT3FE and LT3RE and the semi-inner pair set LT3FI and LT3RE, which amplify a fragment of 151 bp (Table 2) [4]. The amplifications were carried out in a total volume of 50 µL containing 5 µL of DNA template, 500 nM each primer, 0.4 mM dNTPs, 2 mM and 1 mM MgCl_2_ for the outer and the inner amplification, respectively, and 2 U of Taq DNA Polymerase (Solis BioDyne, Tartu, Estonia) in the presence of 10× Reaction Buffer supplied by the manufacturer. The amplification protocols were as follows: an initial denaturation at 95 °C for 5 min, followed by 30 cycles of 30 s denaturation at 95 °C, 30 s annealing at 55 °C for the external amplifications and 50 °C and 59 °C for the internal amplifications with the primer sets FwI-RvI and LT3FI-LT3RE, respectively, and 30 s extension at 72 °C, followed by a final extension step at 72 °C for 5 min. Amplification products were analyzed on a 2% agarose gel in 0.5× TBE.

### 2.6. Statistical Analysis

Count and percentage were used for qualitative variables. The number of samples positive for one HPyV/STI over the total number of collected samples was used to calculate the prevalence of each type of HPyV/STI infection in the cohort of studies. The presence of at least two HPyVs/STI or HPyVs and STIs in patients’ samples was used to calculate the percentage of patients with coinfection. Comparisons between different types of specimens were performed using the chi-square test or Fisher’s exact test. Unpaired t-tests were performed to evaluate the differences among the viral loads. The results were considered significant when the *p* values were < 0.05.

## 3. Results

### 3.1. HPyV and STI Prevalence among Patients

All patients were clinically asymptomatic, but four showed clinically severe symptomatic conditions. The STI most frequently detected was *C. trachomatis* (31/126, 24.6%), followed by *M. genitalium* (30/126, 23.8%) and *N. gonorrhoeae* (4/126, 3.1%). One/126 (0.8%) patient was positive for HPV, while 31/126 (24.6%) patients were positive for at least one HPyV (Table 3).

Among HPyVs, either BKPyV, MCPyV or HPyV6 were found in the clinical specimens, whereas JCPyV, HPyV-7 and HPyV-9 DNA were not amplified in any patient’s sample. The significantly (*p* < 0.05) most prevalent HPyV was MCPyV, which was amplified in 28/126 (23.0%) patients, followed by HPyV-6, which was amplified in 5/126 (4.8%) patients, and by BKPyV, which was found in 1/126 (0.8%) patient (Table 4). Two/126 (1.6%) patients showed the presence of MCPyV in both throat and anal swabs. The four symptomatic patients were MCPyV positive.

### 3.2. STI and HPyV Detection in Clinical Specimens

Analyzing the distribution of the STIs among the different clinical specimens, *C. trachomatis* and *M. genitalium* sequences were most frequently detected in urethral swabs (50% and 33.3%, respectively). Regarding HPyVs, 40/158 (25.3%) samples were positive for at least one HPyV. The HPyV genomes were mostly amplified (*p* < 0.05) in the anal/rectal swab samples (27/81, 33.3%) and in the throat/oral swab samples (12/65, 18.5%), as described in Table 5.

The distribution of HPyVs in the different types of swab samples is summarized in Table 6. MCPyV was the significantly (*p* < 0.05) prevalent HPyV, being amplified in 33/158 (20.9%) samples, followed by HPyV6, isolated in 6/158 (3.8%) samples, and by BKPyV, present in 1/158 (0.6%) sample.

Four samples were coinfected with 2 HPyVs and distributed as follows: 3/4 with MCPyV/HPyV-6 were detected in anal samples, while the remaining one was coinfected with MCPyV/BKPyV in a throat sample.

### 3.3. STI and HPyV Coinfection

The coinfection rates of HPyVs with other ST microorganisms are detailed in Table 7 and stratified by type of sample.

MCPyV sequences were more frequently detected in anal/rectal samples coinfected with *C. trachomatis* (4/21, 19.0%) and *M. genitalium* (5/21, 23.8%). Specifically, MCPyV and *C. trachomatis* (4/21, 19.0%) or *M. genitalium* (5/21, 23.8%) coinfections were significantly detected in anal/rectal samples. Notably, 3/4 symptomatic patients were MCPyV-*C. trachomatis* coinfected. Among the 6 anal/rectal swabs positive for the presence of HPyV-6, 2 were coinfected with *C. trachomatis* and 3 with *M. genitalium*.

### 3.4. HPyV Load

The mean HPyV loads were between 3.77 × 10^5^ copies/mL (range 1.6 × 10^3^–3.8 × 10^6^ copies/mL) for MCPyV and 77 copies/mL for BKPyV. The mean HPyV6 load was 4.2 × 10^4^ copies/mL (range 8.3 × 10^3^–7.6 × 10^4^ copies/mL). The mean percentage of infected cells was 130.5% for MCPyV, 0.01% for BKPyV and 0.90% for HPyV-6. Analytical description of the loads and percentage of infected cells in the different clinical specimens are summarized in Table 8.

### 3.5. MCPyV Molecular Characterization

The automatic sequencing analysis of the amplified LT fragments confirmed that the amplicons belong to the MCPyV genome. In particular, the amplicons showed the highest homology scores with the strain M-cg-9-11.3 (GenBank: MG241580.1), M-cg-28.8.16 (GenBank: MG241582.1), JPN-SK98 (GenBank: LC097004.1), and MCPyV-124 Sweden (GenBank: KX827417.1), with no difference in distribution among the different types of samples. Notably, among symptomatic patients, the strain JPN-SK98 was mostly detected, although without statistical significance (data not shown).

## 4. Discussion

Studies on the prevalence of viral infections in the anal canal of patients are very rare and are mainly focused on HPV infections.

In our study, HPV prevalence was very low compared to all the analogous published studies [20,21,22]. However, rates of anal HPV have been consistently observed to be higher in HIV-positive men compared to HIV-negative men, while our population consists only of HIV-negative subjects [23,24,25,26].

HPV has been detected in over 90% of anal carcinoma biopsy specimens from MSM, and the standardized incidence ratio of anal cancer reported in patients with HIV was 28.75, while patients immunosuppressed after organ transplant had an incidence of 4.85, confirming that HIV is strongly associated with HPV infection and the development of anal cancer [27,28,29].

However, HPV is considered a necessary but independently insufficient factor for carcinogenesis [30,31]. The fact that anal carcinoma can develop even in the absence of HPV infection leads to the hypothesis that other risk factors, including infectious agents, enhance the pathogenetic outcome. Among these, HPyVs assume an increased interest due to their carcinogenic potential and their possible transmission via sexual intercourse.

To date, only two manuscripts reported results from HPyV screenings in MSM [15]. The first, conducted by Peng and colleagues, showed the presence of the genome of at least one HPyV in the anal swabs of 34.2% of the HIV-negative population studied, with the MCPyV genome being isolated in 24.3% of the patients [15]. In the second, Wieland and colleagues detected MCPyV DNA in 30% of normal anal mucosal samples, and in 26% of anal intraepithelial neoplasia samples [32].

In our population, a slightly different general prevalence of HPyV but a very similar prevalence of MCPyV was observed, confirming that the virus is widespread in the adult population [32]. Sequence analysis confirmed the high homology of the strains amplified in our population with strains previously isolated from the healthy skin of the worldwide population [33,34].

To date, there is no evidence of any association between MCPyV infection and anal intraepithelial neoplasia development or other anal tumors; on the contrary, Wieland and colleagues observed a higher frequency of MCPyV infection in normal anal tissue compared to AIN tissues [32]. However, the oncogenic properties of MCPyV, the etiological agent of Merkel cell carcinoma (MCC), a rare and aggressive neoplasia of the skin, [4] are well defined, showing a carcinogenic model similar to that of HPV. Interestingly, the mean percentage of MCPyV-infected cells in the anal swabs of our series was very high, reaching values of up to 1.94 MCPyV copies per β-globin gene copy, but it is also true that a very high variability in viral load was observed. In this regard, it has been assessed that a number between 0.06 and 1.2 viral copies per cell would be sufficient to contribute to neoplasia in MCC cases and that a high abundance of the viral genome is associated with the loss of p53 and pRb activity [35].

Lower MCPyV prevalence and lower percentage of infected cells were found in the throat/oral samples compared to anal swabs. The MCPyV prevalence is in line with what we found in a previous study focused on tonsil specimens [36] and with a recently published study [37] that showed that the oral cavity may represent a latent site of the virus, rather than a site of active replication.

In this regard, Vergori and colleagues speculated that because of differences in local mucosal immunity, nongenital site infections are less likely to reactivate from latency when compared to genital infections, where microtraumatisms and the occurrence of sexually transmitted diseases may decrease the ability of the host to clear viruses [20].

Regarding the other HPyVs, coinciding with the findings of Peng and colleagues, we found that HPyV-6 was the second most prevalent HPyV in the anal swabs. HPyV-6 was first discovered from human skin swabs from healthy volunteers [5], but since then, it has been detected in tonsillar tissue, cerebrospinal fluid, human bile, respiratory tract, feces, and, interestingly, in the cervical mucosa, as part of the normal virome [38]. For MCPyV, a high viral load has been associated with diseases, such as pruritic and dyskeratotic dermatitis, whereas asymptomatic infections have been characterized by a viral load several orders of magnitude lower [39]. Thus, it is likely that in our population, HPyV-6 is present but does not play any relevant pathogenetic role.

Our study also aimed to evaluate HPyV coinfections with the most common worldwide STIs among MSM, including *C. trachomatis, N. gonorrhoeae, Mycoplasmas* and HPV [40,41].

In our MSM study population, a high prevalence of *C. trachomatis* has been observed, as well as a high prevalence of *M. genitalium* both in rectal and pharyngeal sites. Previously, persistent *C. trachomatis*, recognized as an event with oncogenic potential and often found in samples coinfected with HPV, was reported in this geographic area in urethra samples from heterosexual men at risk for STIs [42]. Regarding *Mycoplasmas*, although frequent infections have been reported in sperm from men affected by infertility [43], controversial opinions on their role as colonizers or pathogens have been discussed. On the other hand, their ability to induce long-lasting inflammation and tissue damage at the site of infection [44,45] is emphasized.

Our study did not include any control group, and this is due to the fact that, so far, there are no recommendations for performing routine screening of the general population for anal infection and/or cancer. In a recent review, Leeds and Fang reported a list of National Associations, comprising the CDC and the British HIV association, that give no recommendations for anal screening in men who have sex with women (MSW) [46]. Based on this observation, it is hard to obtain IRB approval from any local ethic committee to collect anal swabs from no-risk subjects. The almost completely lack of data regarding anal infection in MSW, in the scientific literature, confirms the difficult in obtaining this type of clinical specimens. So far, there are only seven published manuscripts on the detection of HPV in the anal canal of HIV-negative men practicing sex with women (MSW), and two regarding HIV-positive MSW, most from the Americas (Brazil, Mexico, United States and Puerto-Rico), one from Asia (Korea), and one from Russia. Additionally, five out of ten studies were conducted by the same work group [47,48,49,50,51,52,53,54,55]. However, none of these publications is about HPyVs, but they showed only the high prevalence (33.9%) of any HPV in heterosexual men (Table 9); the difference of HPV prevalence among these studies and ours may be due to the different analyzed geographical areas.

Interestingly, in our study, *C. trachomatis* and *M. genitalium* were the unique STI associated with MCPyV in the same samples. Symptomatic patients were predominantly infected with MCPyV and *C. trachomatis,* and to the best of our knowledge, this is the first description of this coinfection. We hypothesize that a long-lasting asymptomatic infection of the anal mucosa with *C. trachomatis* represents an extremely favorable event for MCPyV acquisition and the possible development of severe lesions in the presence of a high viral load. The high prevalence of MCPyV in asymptomatic patients, similar to that observed for *C. trachomatis*, could represent a new challenge in the field of STIs.

## 5. Conclusions

This is the first study on HPyV prevalence and other STIs in an Italian group of MSM. MCPyV prevalence was similar to that already described in other geographical areas, and its widespread diffusion was confirmed. However, based on the high prevalence of MCPyV in anal swabs, it is strongly suggested that further studies be conducted to confirm the sexual transmission route of this virus and other HPyVs, as well as their role in the severity of clinical outcome.

In particular, further evaluation of MCPyV is suggested, especially in association with other ST pathogens and demographic risk factors, as it could be responsible for cell transformation of the anal tract based on its well-known oncogenic abilities.

## Figures and Tables

**Table 1 microorganisms-07-00054-t001:** Primer and probe sequences for HPyVs.

Virus	Target Region	Nucleotide Numbering	Forward Primer Reverse Primer Probe	Concentration µM
JCV ^1^	LT	4299–4321	5′-GAGTGTTGGGATCCTGTGTTTTC-3′	0.4
		4352–4375	5′-GAGAAGTGGGATGAAGACCTGTTT-3′	0.4
		4323–4350	5′-FAM-TCATCACTGGCAAACATTTCTTCATGGC-MGB-3′	0.15
BKPyV ^2^	VP1	2511–2531	5′-AGTGGATGGGCAGCCTATGTA-3′	0.2
		2586–2605	5′-TCATATCTGGGTCCCCTGGA-3′	0.4
		2535–2556	5′-VIC-TATGGAATCCCAGGTAGAAGA-MGB-3′	0.2
MCPyV ^3^	VP1	4053–4072	5′-TGCCTCCCACATCTGCAAT-3′	0.2
		4090–4112	5′-GTGTCTCTGCCAATGCTAAATGA-3′	0.4
		4074–4089	5′-FAM-TGTCACAGGTAATATC-MGB-3′	0.15
HPyV-6 ^4^	VP1	1767–1786	5′-GGCCTGGAAGGGCCTAGTAA-3′	0.9
		1847–1823	5′-ATTGGCAGCTGTAACTTGTTTTCTG-3′	0.9
		1789–1806	5′-JOE-AGAACCAACCATCTGTTG-BHQ1-3′	0.25
HPyV-7 ^5^	VP1	1774–1796	5′-AGGTCAATGAAGCCCTAGAAGGT-3′	0.2
		1840–1822	5′-TGCTTTCTGAGGGCTTGCA-3′	0.9
		1798–1817	5′-FAM-CAGGCAATACTGATGTAGC-MGB-3′	0.15
HPyV-9 ^6^	VP1	1449–1469	5′-CCCCAAAGAAAAGGCAAGAG-3′	0.4
		1509–1493	5′-GCGGGTGTTGGACAGGTTT-3′	0.9
		1472–1488	5′-VIC-CGGAGCATGTCCTGTAA-MGB-3′	0.25

^1^ Strain MAD 1 (GenBank accession number J02226.1); ^2^ Strain WW (GenBank accession number AB211371); ^3^ Isolate MCC 350 (GenBank accession number EU375803); ^4^ Complete genome (GenBank accession number HM011563); ^5^ Complete genome (GenBank accession number HM011566); ^6^ Complete genome (GenBank accession number HQ696595).

**Table 2 microorganisms-07-00054-t002:** Primer sequences for MCPyV LTAg.

Outer/Inner PCR	Primers	Nucleotide Numbering *	Sequence	Fragment Length
Outer	Tag FwE	1716–1736	5′-GCCTGTGAATTAGGATGTATTTT-3′	477 bp
	Tag RevE	2210–2198	5′-TGGAAATGACCAGGACAGAAATG-3′	
Inner	Tag FwI	2010–2033	5′-GCCCATTATCTAGACTTTGCAAA-3′	183 bp
	Tag RevI	2192–2173	5′-TAGCCAAAAGGAGGTTAGA-3′	
Outer	LT3FE	571–590	5′-TTGTCTCGCCAGCATTGTAG-3′	280 bp
	LT3RE	860–879	5′-ATATAGGGGCCTCGTCAACC-3′	
Inner	LT3FI	741–760	5′-ATCTGCACCTTTTCTAGACTCC-3′	151 bp
	LT3RE	860–879	5′-ATATAGGGGCCTCGTCAACC-3′	

* Nucleotide position refers to strain EU375803.

**Table 3 microorganisms-07-00054-t003:** STI prevalence among patients.

Sexually Transmitted Infections					
Patients (*n*)	HPyV+/Tot (%)	HPV +/Tot (%)	*C. trachomatis* +/Tot (%)	*M. genitalium* +/Tot (%)	*N. gonorrhoeae* +/Tot (%)
126	31/126 * (24.6)	1/126 *^,^** (0.8)	31/126 ** (24.6)	30/126 (23.8)	4/126 *^,^** (3.1)

*= *p* < 0.0001; ** = *p* ≤ 0.001, Tot: Total.

**Table 4 microorganisms-07-00054-t004:** HPyV prevalence among patients.

HPyVs			
Patients (*n*)	BKPyV+/Tot *^,^** (%)	MCPyV+/Tot ** (%)	HPyV-6+/Tot * (%)
126	1/126 *^,^** (0.8)	28/126 *^,^** (23.0)	5/126 * (4.8)

*= *p* < 0.0001; ** = *p* ≤ 0.001, Tot: Total.

**Table 5 microorganisms-07-00054-t005:** STI and HPyV distribution among samples.

Sample	*n*	HPV+/Tot (%)	HPyV+/Tot (%)	*C. trachomatis*+/Tot (%)	*M. genitalium* +/Tot (%)	*N. gonorrhoeae* +/Tot (%)
Anal/rectal	81	1/81 (1.2)	27/81 (33.3)	22/81 (27.1)	21/81 (26)	3/81 (3.7)
Throat/oral	65	0/65 (0.0)	12/65 (18.4)	10/65 (15.3)	8/65 (12.3)	3/65 (4.6)
Urethral	12	0/12 (0.0)	1/12 (8.3)	6/12 (50)	4/12 (33.3)	0/12 (0.0)
Total	158	1/158 (0.6)	40/158 (25.3)	38/158 (24)	33/158 (20)	6/158 (3.8)

**Table 6 microorganisms-07-00054-t006:** Distribution of HPyVs in the clinical specimens.

Sample	*n*	BKPyV+/Tot (%)	MCPyV+/Tot (%)	HPyV-6+/Tot (%)
**Throat/Oral**	65	1/65 (1.5)	11/65 (16.9)	0/65 (0.0)
**Anal/Rectal**	81	0/81 (0.0)	21/81 (25.9)	6/81 (7.4%)
**Urethral**	12	0/12 (0.0)	1/12 (8.3)	0/12 (0.0)
**Total**	158	1/158 (0.6) *	33/158 (20.9) *^,^**	6/158 (3.8) **

*= *p* < 0.0001; ** = *p* ≤ 0.001, Tot: Total.

**Table 7 microorganisms-07-00054-t007:** Distribution of HPyVs in samples coinfected with ST microorganisms.

STI	MCPyV+		BKPyV+		HPyV-6+	
	Anal/Rectal (%)	Throat/Oral (%)	Anal/Rectal	Throat/Oral (%)	Anal/Rectal (%)	Throat/Oral
***C. trachomatis***	4/21 (19.0)	1/11 (9.1)	0/0	0/1	2/6 (33.3)	0/0
***M. genitalium***	5/21 (23.8)	1/11 (9.1)	0/0	1/1 (100)	3/6 50.0)	0/0
***N. gonorrhoeae***	0/21	1/11 (9.1)	0/0	0/1	0/6	0/0
**HPV**	1/21 (9.1)	0/11	0/0	0/1	0/6	0/0

**Table 8 microorganisms-07-00054-t008:** The mean viral load and percentage of HPyV-infected cells in the clinical specimens.

Sample	BKPyV		MCPyV		HPyV-6	
	% Infected Cell (Range)	Mean Viral Load Copies/mL (Range)	% Infected Cell (Range)	Mean Viral Load Copies/mL (Range)	% Infected Cell (Range)	Mean Viral Load Copies/mL (Range)
**Throat/Oral**	0.01%	77	8.7% (0.03%–68.9%)	3.6 × 10^4^ (8.2 × 10^3^–1.5 × 10^5^)	-	**-**
**Anal/Rectal**	-	-	194.3% (0.1%–3377.0%)	5.4 × 10^5^ (1.6 × 10^3^–3.8 × 10^6^)	0.90% (0.01%-4.1%)	4.2 × 10^4^ (8.3 × 10^3^–7.6 × 10^4^)
**Urethral**	-	-	0.34%	2.5 × 10^3^	-	-
Total	0.01%	77	130.5% (0.1%–3377.0%)	3.77 × 10^5^ (1.6 × 10^3^–3.8 × 10^6^)	0.90% (0.01%–4.1%)	4.2 × 10^4^ (8.3 × 10^3^–7.6 × 10^4^)

**Table 9 microorganisms-07-00054-t009:** HPV anal tract infection in MSW, with no risk factors

Reference	HPV+/Total	%
[47]	68/386	17.6
[48]	36/222	16.6
[49]	108/902	12.0%
[50]	159/1305	12.2%
[51]	1766/3326	53.1%
[52]	58/144	40.3%
[55]	52/332	15.7%
**Total**	**2247/6617**	**33.9%**

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
