# Peer review of "Merkel Cell Polyomavirus Is Associated with Anal Infections in Men Who Have Sex with Men"

_microorganisms, 2019, doi:10.3390/microorganisms7020054_

Round 1

Reviewer 1 Report

Zanotti and colleagues analyze the presence of sexually transmitted pathogens including Chlamydia trachomatis, Neisseria gonorrhoeae and Mycoplasma in 158 swabs from 126 patients (anal/rectal, throat/oral and urethral) of men who have sex with men. Furthermore, the presence of human papillomaviruses and human polyomaviruses (BKV, JCV, MCPyV, HPyV6, 7 and 9) was analyzed in these swabs. The conclusion of the authors based on high prevalence (33.3%) of MCPyV in anal/rectal swabs is that further studies are needed to deepen the hypothesis that MCPyV due to its oncogenic properties might contribute to pathogenesis of the anal canal.

I have severe problems following this conclusion. The authors did not include a control group, neither did they analyze skin swabs of the patients. MCPyV is highly prevalent and can be detected on the skin and mucosal skin of healthy adults. The presence of the virus on the skin or maybe also in anal/rectal swabs is not indicative of a causative contribution to a pathogenesis.

Is there any indication that MCPyV can be detected in AIN and/or anal cancer in general and in the MSM group in particular?

Besides this, the manuscripts needs significant improvement of the language. Furthermore, the introduction needs some evaluation: in particular with regard to the background knowledge provided for PyVs. Some of the statements are only true for JC or BKV (line 41, persistence: lymphoid tissue is most likely the site or one site of persistence for JC Virus while BK Virus persist in the renal epithelium. Prevalence of hPyV are not 100% (line 39) BK Virus has the highest prevalence with approximately 90% in patients older 70 years of age.

The high prevalence of MCPyV in mucosal swabs (30%) in a study performed by Wieland et al., 2009 need to be discussed.

Minor points:

Line 34: naked should be replaced by non-enveloped

Primer sequences, probe sequences and PCR conditions for all hPyV need to be described.

Which type of Mycoplasma were detected (line 73)

Line 95: count and percentages

Author Response

Zanotti and colleagues analyze the presence of sexually transmitted pathogens including Chlamydia trachomatis, Neisseria gonorrhoeae and Mycoplasma in 158 swabs from 126 patients (anal/rectal, throat/oral and urethral) of men who have sex with men. Furthermore, the presence of human papillomaviruses and human polyomaviruses (BKV, JCV, MCPyV, HPyV6, 7 and 9) was analyzed in these swabs. The conclusion of the authors based on high prevalence (33.3%) of MCPyV in anal/rectal swabs is that further studies are needed to deepen the hypothesis that MCPyV due to its oncogenic properties might contribute to pathogenesis of the anal canal.

I have severe problems following this conclusion. The authors did not include a control group, neither did they analyze skin swabs of the patients. MCPyV is highly prevalent and can be detected on the skin and mucosal skin of healthy adults. The presence of the virus on the skin or maybe also in anal/rectal swabs is not indicative of a causative contribution to a pathogenesis.

Is there any indication that MCPyV can be detected in AIN and/or anal cancer in general and in the MSM group in particular?

To date, only two papers reported results on the presence of MCPyV genome in anal samples , Peng et al., 2016 and Wieland et al., 2009. While the first one did not refer to patients with AIN and/or anal cancer, the second one showed that there is no association between presence of MCPyV genome and AIN. We added this information in the discussion section.

Besides this, the manuscripts needs significant improvement of the language.

The manuscript was edited by the American Journal Expert and the certificate was uploaded.

Furthermore, the introduction needs some evaluation: in particular with regard to the background knowledge provided for PyVs. Some of the statements are only true for JC or BKV (line 41, persistence: lymphoid tissue is most likely the site or one site of persistence for JC Virus while BK Virus persist in the renal epithelium. Prevalence of hPyV are not 100% (line 39) BK Virus has the highest prevalence with approximately 90% in patients older 70 years of age.

All the suggested corrections were made in the Introduction section.

The high prevalence of MCPyV in mucosal swabs (30%) in a study performed by Wieland et al., 2009 need to be discussed.

The results obtained by Wieland and colleagues were discussed in the discussion section.

Line 34: naked should be replaced by non-enveloped

Primer sequences, probe sequences and PCR conditions for all hPyV need to be described.

Which type of Mycoplasma were detected (line 73)

Line 95: count and percentages

All the suggested corrections were made and a table was added for describing primer and probe sequences.

Reviewer 2 Report

Zanotta et al. investigated the presence of DNA from the human papillomaviruses (HPV) and the polyomaviruses MCPyV, HPyV6, HPyV7 and HPyV9 in swab samples (anal, oral and urethral) of men who have sex with men. All individuals were HIV-negative and clinical symptomatic except for four who showed pruritic and dyskeratotic dermatitis. The presence of Chlamydia trachomatis, Neisseria gonorrhoeae and Mycoplasmas was also examined. Real time PCR was used. The authors found that MCPyV was the most prevalent human polyomavirus, followed by HPyV6. Only one patient had a HPV positive sample, while C. trachomatis was most frequently detected, followed by Mycoplasmas and N. gonorrhoea. Co-infection was detected in some patients. The authors conclude that because of the oncogenic properties and the relatively high prevalence (33%) of MCPyV in anal swab samples, the possible involvement in MCPyV in pathogenic diseases of the anal canal needs further attention.

This is a well-performed study on a relative large number of samples. Statistical analyses were used to establish significant differences or co-infections.

Minor comments

Line 36: there are currently 14 polyomaviruses that have been detected in humans. Gheit et al. described a novel one in their Virology paper of 2017.

Line 39: reference 6 is incorrect. It refers to TSPyV and not HPyV9. The correct reference is Scuda N et al., 2011.

Line 39: the authors should specify that the numbers 60% to 100% refer to seropositivity.

Line 43-44: because the authors focus on MCPyV, HPyV6 and HPyV7, they should mention that these viruses are common in the skin.

Lines 58-66: the authors should mention here that none of the patients was HIV positive. This information is not provided until the reader reaches the discussion (line 156).

Line 142: The authors should add that the BKPyV positive urethral sample had no co-infection with Chlamydia trachomatis, Neisseria gonorrhoeae and Mycoplasmas.

Discusion, lines 171-180: Hallmarks for MCPyV-positive Merkel cell carcinoma tumors are the integration of the viral genome and expression of a truncated Large T-antigen. Have the authors examined the status of the MCPyV genome in the mucosa cells of anal swabs and sequenced the Large T-antigen? If not, it would be interesting to examine this to establish whether MCPyV-induced anal cancer (in case MCPyV could be involved in anal malignancies) uses the same mechanism as in virus-positive MCC.

English can be improved and some typos. E.g.:

                -line 53: based on these few information

                -line 173: to that of HPV.

                -lines 208 and 210: thracomatis (no capital t)

                -lines 219 and 222: HPyV

                -line 221: Additionally, in the light of its oncogenic properties and its similarities with HPV

                -Table 3 and Table 5 on page 5: gonorrhoeae

Author Response

Zanotta et al. investigated the presence of DNA from the human papillomaviruses (HPV) and the polyomaviruses MCPyV, HPyV6, HPyV7 and HPyV9 in swab samples (anal, oral and urethral) of men who have sex with men. All individuals were HIV-negative and clinical symptomatic except for four who showed pruritic and dyskeratotic dermatitis. The presence of Chlamydia trachomatis, Neisseria gonorrhoeae and Mycoplasmas was also examined. Real time PCR was used. The authors found that MCPyV was the most prevalent human polyomavirus, followed by HPyV6. Only one patient had a HPV positive sample, while C. trachomatis was most frequently detected, followed by Mycoplasmas and N. gonorrhoea. Co-infection was detected in some patients. The authors conclude that because of the oncogenic properties and the relatively high prevalence (33%) of MCPyV in anal swab samples, the possible involvement in MCPyV in pathogenic diseases of the anal canal needs further attention.

This is a well-performed study on a relative large number of samples. Statistical analyses were used to establish significant differences or co-infections.

Minor comments

-          Line 36: there are currently 14 polyomaviruses that have been detected in humans. Gheit et al. described a novel one in their Virology paper of 2017.

The correction was made and the reference was added.

-          Line 39: reference 6 is incorrect. It refers to TSPyV and not HPyV9. The correct reference is Scuda N et al., 2011.

The reference was changed.

-          Line 39: the authors should specify that the numbers 60% to 100% refer to seropositivity.

-          Line 43-44: because the authors focus on MCPyV, HPyV6 and HPyV7, they should mention that these viruses are common in the skin.

-          Lines 58-66: the authors should mention here that none of the patients was HIV positive. This information is not provided until the reader reaches the discussion (line 156).

All the corrections were made.

-          Line 142: The authors should add that the BKPyV positive urethral sample had no co-infection with Chlamydia trachomatis, Neisseria gonorrhoeae and Mycoplasmas.

We could not make this correction, since no BKPyV positive urethral sample has been found in the study. BKPyV genome was only found in an oral/throat sample, that was coinfected with C. trachomatis.

-          Discussion, lines 171-180: Hallmarks for MCPyV-positive Merkel cell carcinoma tumors are the integration of the viral genome and expression of a truncated Large T-antigen. Have the authors examined the status of the MCPyV genome in the mucosa cells of anal swabs and sequenced the Large T-antigen? If not, it would be interesting to examine this to establish whether MCPyV-induced anal cancer (in case MCPyV could be involved in anal malignancies) uses the same mechanism as in virus-positive MCC.

Unfortunately, we could not examined the status of the MCPyV genome, since small amount of DNA from the samples was left. Instead, we perform the sequencing analysis of a fragment of the LT gene, in order to verify which type of MCPyV infected the anal tract of the patients. We added the methods and the results.

English can be improved and some typos. E.g.:

                -line 53: based on these few information

                -line 173: to that of HPV.

                -lines 208 and 210: thracomatis (no capital t)

                -lines 219 and 222: HPyV

                -line 221: Additionally, in the light of its oncogenic properties and its similarities with HPV

                -Table 3 and Table 5 on page 5: gonorrhoeae

English language was edited by the American Journal Expert and the certificate of editing was uploaded.

Round 2

Reviewer 1 Report

The authors improved all minor points raised by the reviewers. However, the major concern with the manuscript, in particular with the conclusion of the manuscript remains.

The presence of the virus on the skin or in anal/rectal swabs is not indicative of a causative contribution to a pathogenesis/neoplasia. Furthermore, DNA copy measurements (which are very often highly variable) are not sufficient to conclude pathogenesis/transforming potential of MCPyV. Here, integration of the viral DNA is essential to show. Alternatively, the truncating mutation/deletion in the early region of the virus has to be shown. Saying this, the title of the manuscript and the discussion part line 240 – 245 are not supported by the data.

Still it remains that the authors did not include a control group.

Author Response

Dear editor,

thank you for taking into consideration the revised paper.

Unfortunately, although we agree with you and believe that it would be of great interest, we are not able to provide data regarding the HPyVs and ST pathogens prevalence in a control group consisting of Men who have sex only with woman (MSW), for several reasons, that were also added in the discussion session.

- So far, there are no recommendations for performing routine screening of general population for anal infection and/or cancer. In a recent review, Leeds and Fang reported a list of National Associations, comprising the CDC and the British HIV association, that give no recommendations for the anal screening in MSW, but that, of course, indicated that Men who have sex with men, women with a history of abnormal cervical Pap tests, and all HIV-positive persons with genital warts should undergo anal screening Leeds IL, Fang SH. Anal cancer and intraepithelial neoplasia screening: A review. (World J Gastrointest Surg. 2016 Jan 27;8(1):41-51).

- Based on this observation, it is almost impossible to obtain the IRB approval from any local ethic committee for collecting anal swabs from no risk subjects.

- The almost completely lack of data regarding anal infection in MSW, in the scientific literature, confirms the difficult in obtaining this type of clinical specimens. So far, there are only 9 published manuscripts on the detection of the HPV in the anal canal of HIVnegative men practicing sex with women (MSW), most from the Americas (Brazil, Mexico, United States and PuertoRico), one from Asia (Korea), and one from Russia. Additionally, five out of ten studies were conducted by the same work Group (Giuliano et al., 2007; Nyitray et al., 2008; 2010; 2011a and b; Colon Lopez et al., 2014; Melo et al., 2014 and Lee et al., 2016 and  Smelov et al., 2018).

Hoping that you will understand our issue, and hoping in your positive evaluation

I send you

Best regards

Pasquale Ferrante
